# RaySt3R: Predicting Novel Depth Maps for Zero-Shot Object Completion

**Bardienus P. Duisterhof**
Carnegie Mellon University

**Jan Oberst**
Carnegie Mellon University

**Bowen Wen**
NVIDIA

**Stan Birchfield**
NVIDIA

**Deva Ramanan**
Carnegie Mellon University

**Jeffrey Ichnowski**
Carnegie Mellon University

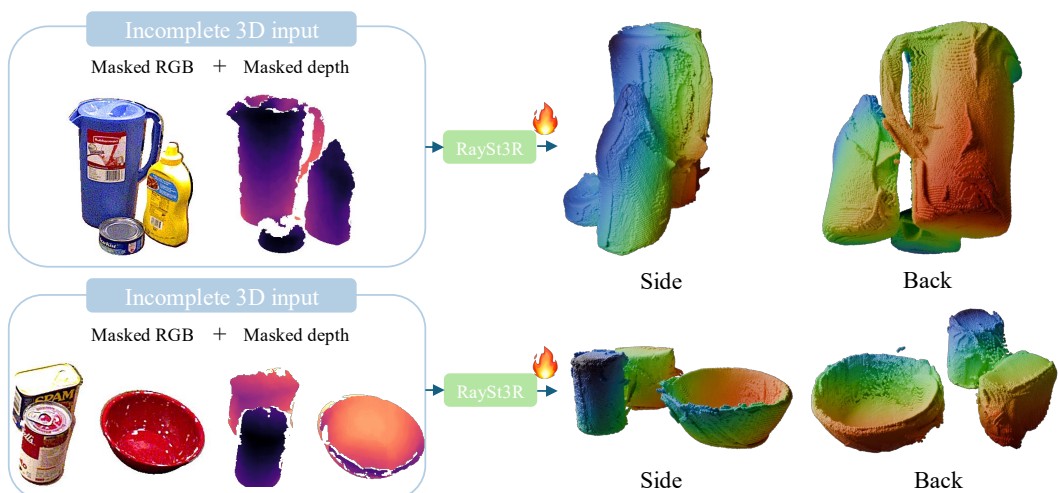

Figure 1: RaySt3R is a method for zero-shot 3D shape completion from a single foreground-masked RGB-D image. It predicts depth maps, object masks, and per-pixel confidence scores for novel viewpoints, and fuses them to reconstruct a complete 3D shape. RaySt3R is able to recover the geometry of full objects in cluttered real-world scenes, despite only being trained on synthetic data.

## Abstract

3D shape completion has broad applications in robotics, digital twin reconstruction, and extended reality (XR). Although recent advances in 3D object and scene completion have achieved impressive results, existing methods lack 3D consistency, are computationally expensive, and struggle to capture sharp object boundaries. Our work (RaySt3R) addresses these limitations by recasting 3D shape completion as a novel view synthesis problem. Specifically, given a single RGB-D image and a novel viewpoint (encoded as a collection of query rays), we train a feedforward transformer to predict depth maps, object masks, and per-pixel confidence scores for those query rays. RaySt3R fuses these predictions across multiple query views to reconstruct complete 3D shapes. We evaluate RaySt3R on synthetic and real-world datasets, and observe it achieves state-of-the-art performance, outperforming the baselines on all datasets by up to 44 % in 3D chamfer distance. Project page: `rayst3r.github.io`

39th Conference on Neural Information Processing Systems (NeurIPS 2025).

# 1 Introduction

3D shape completion is an enabling tool for visual reasoning and physical interaction with partially visible objects, and facilitates a wide range of downstream tasks such as robot grasping in cluttered environments [29, 50], obstacle avoidance [32, 41], mechanical search [17], digital-twin reconstruction, and Augmented or Virtual or Extended Reality (AR/VR/XR) applications.

**Challenges.** We focus on the robotics-driven setting where an RGB-D image is provided as input for multi-object shape completion. While object-centric methods achieve high reconstruction quality for single objects, multi-object scenes require instance segmentation and alignment procedures [1] that tend to be brittle in practice. Generative approaches use 2D image generation models [59, 49] to generate images from novel viewpoints, but these can be sensitive to large viewpoint changes and are computationally inefficient at inference time, which hinders robot and XR deployment. Other methods scale up 3D prediction on abundant synthetic scene data [18], but the resolution for the 3D representations (such as 3D MAE voxel grids) is too coarse to capture sharp object shapes with high-frequency geometry details.

**Approach.** We propose Ray Stereo 3D Reconstruction (RaySt3R), a novel method for addressing the above challenges. Given a single masked RGB-D image as input, our key insight is to recast shape completion as a novel view-synthesis task, then aggregate multiple view predictions to generate a complete 3D shape. Our approach draws inspiration from recent work that casts 3D reconstruction as point map regression via multi-view transformers [48, 9]. We similarly use a vision transformer (ViT) [22] architecture defined over visual DINOv2 [33] features extracted from the input image. However, instead of requiring a second image as an additional input, we input the novel view to be synthesized in the form of a camera ray map. Specifically, RaySt3R is trained to predict depth maps, confidence maps, and foreground masks for each queried ray via cross-attention. We then merge RaySt3R's geometric predictions from multiple novel views using the per-ray confidence and mask predictions.

**Data.** Since RaySt3R can be seen as a view-synthesis engine, we can train at scale on pairs of RGB-D images without requiring volumetric 3D supervision (as required by prior work [18]). We train RaySt3R on a large-scale augmented synthetic dataset with 251 k unique scenes and 11 million novel depth maps (training pairs). Across synthetic and real-world benchmarks, RaySt3R outperforms prior art by up to 44 % (in shape completion accuracy). Despite never being trained on real data, RaySt3R generalizes well to real-world cluttered scenes.

This paper contributes:

- RaySt3R, a method for view-based 3D shape completion that learns confidence-aware depth maps and object masks from novel views, and uses a novel formulation of merging multi-view prediction.
- A new curated large-scale dataset with 11 million novel depth maps and masks, which we will open-source to facilitate future research.
- Evaluations of RaySt3R on synthetic and real-world datasets that show RaySt3R achieves state-of-the-art accuracy for 3D shape completion, and successfully generalizes to real-world cluttered scenes after training on only synthetic data.

# 2 Related work

3D shape completion has seen impressive progress over the last years. We explore related works categorized by their reasoning space, i.e., volumetric approaches and view-based approaches.

**Volumetric reasoning** Volumetric methods operate directly in 3D space and provide strong geometric priors. Some approaches directly predict point clouds [11, 30, 43], while others rely on implicit geometry representations. The latter infer 3D structure at test time by querying the representation with a spatial point and a partial observation (e.g., an RGB or RGB-D scan). Prior work uses signed distance functions for such representations [34, 23] or voxel occupancy grids [3, 4, 35, 16, 31, 53, 24]. OctMAE [18] builds on the idea of MAE [13] from the image synthesis domain, and applies it to next-token prediction natively in 3D. Although these methods yield promising results, their resolution

---

This work was generously supported by the Center for Machine Learning and Health (CMLH) at CMU, the NVIDIA Academic Grant Program, and the Pittsburgh Supercomputing Center.

is constrained by the cubic cost of their volumetric resolution, leading to a coarse grid and smoothed structures lacking fine details.

Another strategy decomposes the scene at the object level [1, 58]. SceneComplete [1] constructs a 3D scene by chaining together foundation models for object segmentation, occlusion inpainting, 3D shape retrieval, and pose estimation. Despite its modular design, our experiments (Section 5) suggest that this reliance on multiple components introduces brittleness and several single points of failure.

Recent work like TRELLIS [55] instead learns a 3D latent representation from text or image, which can be decoded into various formats such as meshes. However, as shown in Section 5, experiments suggest it struggles in real-world multi-object scenes. In contrast, RaySt3R adopts a view-based strategy that is specifically designed for robust 3D completion in cluttered environments.

**View-based reasoning**   Diffusion models [14] and their extensions [15, 39, 38] have enabled unprecedented performance in generative tasks such as image synthesis, inpainting, and video prediction. Several works leverage off-the-shelf generative models for 3D generation tasks [27, 59, 44, 54, 12]. ViewCrafter [59] leverages these advances by iteratively completing a scene point cloud using a point-conditioned video diffusion model. Similar to RaySt3R, LVSM [20] synthesizes novel views by querying a transformer with Plücker Rays and by conditioning on input views. RaySt3R predicts depths and object masks instead of images, and does not require full Plücker rays for querying novel views. RayZer [19] is a self-supervised large view synthesis model, using its self-predicted camera poses to eliminate the need for any ground-truth camera annotations.

In the object-centric domain, Li et al. [25] predicts layered depth maps for constructing object-level and scene-level 3D geometries. While this method yields promising results on single-object shape completion, it struggles with predicting accurate geometries in real-world scenes containing multiple objects. Unique3D [54] tries to strike a balance between fidelity and inference speed by predicting multi-view images, generating corresponding normal maps and a textured mesh within 30 seconds.

While these models often yield visually appealing results, they lack geometric consistency, especially for cluttered real world environments. The inference time of large diffusion models may also hinder deployment in robotics or XR settings. In contrast, RaySt3R predicts geometrically accurate depth maps from novel views, for fast and accurate 3D shape completion in cluttered real-world scenes.

## 3   Problem statement

Given a single RGB-D image, $I^{\text{input}} \in \mathbb{R}^{H \times W \times 3}, D^{\text{input}} \in \mathbb{R}^{H \times W}$, foreground mask $M \in \{0, 1\}^{H \times W}$, and a camera with known intrinsics, $K^{\text{input}} \in \mathbb{R}^{3 \times 3}$, the goal is to predict the full 3D surface geometry of all masked foreground objects. We frame the prediction goal as a set of points $Q \in \mathbb{R}^{N \times 3}$ that is both *accurate* and *complete* w.r.t. the ground-truth points $Q^{\text{gt}} \in \mathbb{R}^{S \times 3}$, sampled on the surfaces (e.g., meshes) of all objects in the scene. We measure accuracy as the shortest distance from a predicted point to the nearest ground-truth point, averaged over all predicted points. We measure completeness as the shortest distance from a ground-truth point to the nearest predicted point, averaged over all ground truth points.

## 4   Methods

An overview of our approach is illustrated in Figure 2. We propose to train a transformer that, given the partial capture from a single RGB-D image and foreground mask, predicts depth maps and per-pixel confidence scores, and foreground masks for novel views. We first present the model architecture (Section 4.1), then the training objectives (Section 4.2). We then describe the procedure for querying novel views (Section 4.3) and conclude with the prediction merging strategy (Section 4.4).

### 4.1   Network architecture

The RaySt3R network architecture is inspired by DUSt3R [48] and successors [47, 49, 45]. Here, we leverage a ViT with point map, ray map, and depth map representations for 3D object completion.

The inputs to RaySt3R (Figure 2) are a foreground-masked RGB-D image and a novel query view. First, we unproject the input depth map $D^{\text{input}}$ to a point map $X^{\text{input}} \in \mathbb{R}^{H \times W \times 3}$ using the given

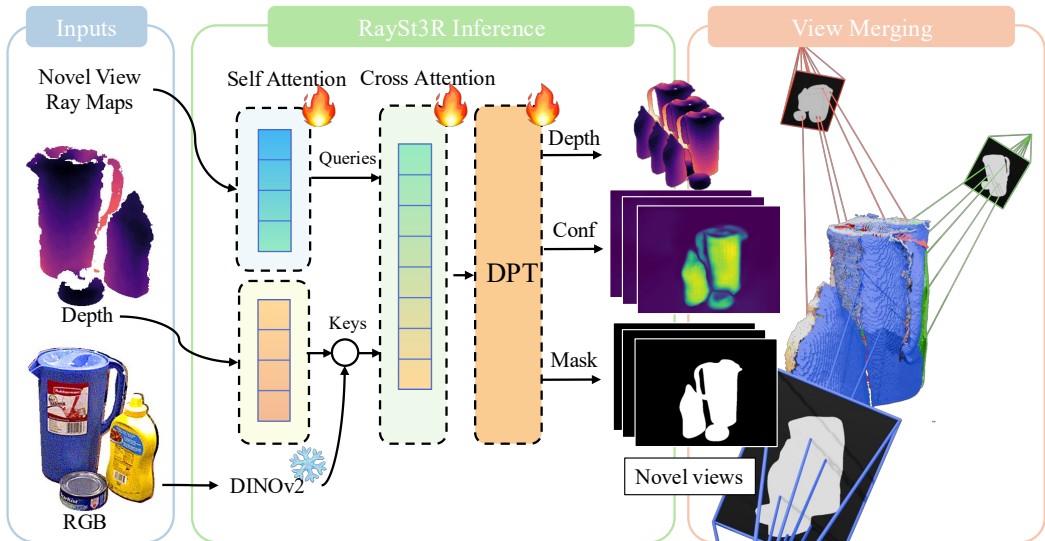

Figure 2: The architecture of RaySt3R. RaySt3R takes a single RGB-D image and foreground mask as input, and predicts depth maps, object masks, and per-pixel confidence scores for novel views. First, we apply the foreground mask to the RGB and point map input. Next, we use self-attention layers for the point map and ray map inputs, and feed the RGB image into the frozen DINOv2 [33] encoder. We feed all features into cross-attention layers followed by two separate DPT heads [36] for depth and mask predictions. Finally, we provide confidence- and occlusion-aware multi-view merging formulation.

input image intrinsics $K^{\text{input}}$, thus $X_{i,j}^{\text{input}} = (K^{\text{input}})^{-1}[iD_{i,j}^{\text{input}}, jD_{i,j}^{\text{input}}, D_{i,j}^{\text{input}}]^{\mathsf{T}}$. We convert the query view into a ray map $R \in \mathbb{R}^{H \times W \times 2}$, with $R_{i,j} = [(i - c_x)/f_x, (j - c_y)/f_y]^{\mathsf{T}}$, where $c_x, c_y$ is the image center and $f_x, f_y$ are the focal lengths of the novel target view.

Because we would like to pass information between the input and query views via cross attention, we transform the input point map $X_{\text{input}}$ into the target camera coordinate frame to ease information sharing. That is, $X^{\text{context}} = P_{\text{input}}^{\text{query}} h(X^{\text{input}})$, where $P_{\text{input}}^{\text{query}} \in \mathbb{R}^{3 \times 4}$ transforms from the input to query camera coordinate frame, and $h : (x, y, z) \mapsto (x, y, z, 1)$.

We compute features for the point map and ray query with $L$ layers of self-attention (SA).

$$F^{\text{point\_map}} = \text{SA}(X^{\text{context}}), \quad F^{\text{ray}} = \text{SA}(R) \tag{1}$$

We mask out the background in $X^{\text{context}}$ and replace it with a single learned background token. We process the RGB image by first masking out the background, and subsequently passing it through a frozen DINOv2 [33] encoder. Recent work has shown that a combination of features from different layers of a pre-trained ViT is useful for downstream tasks [10], hence we concatenate the features from intermediate layers of the DINOv2 encoder, and use a linear layer to project them to $F^{\text{DINO}}$.

For the cross-attention (CA) layers, we construct the keys of the first layer by concatenating $F^{\text{point\_map}}$ and $F^{\text{DINO}}$. The queries are the ray features $F^{\text{ray}}$.

$$G = \text{CA}(F^{\text{ray}}, \text{concat}(F^{\text{point\_map}}, F^{\text{DINO}})) \tag{2}$$

Finally, we use a DPT head [36] to predict depth maps, and its confidence scores. A separate DPT head predicts the object mask.

## 4.2 Training objectives

We train RaySt3R to predict confidence-aware depth maps and object masks.

**Depth loss** Inspired by DUSt3R [48], we define the confidence-aware depth loss:

$$\mathcal{L}_{\text{depth}} = \sum_{i \in [0, W-1]} \sum_{j \in [0, H-1]} M_{i,j}^{\text{gt}} \left( C_{i,j} \left\| d_{i,j} - d_{i,j}^{\text{gt}} \right\|_2 - \alpha \log C_{i,j} \right). \tag{3}$$

Here, $C_{i,j}$ is the confidence score of each pixel in the predicted depth map, $\alpha$ is a hyper parameter, $d_{i,j}$ is the predicted depth, $d_{i,j}^{\text{gt}}$ is the ground-truth depth, and $M_{i,j}^{\text{gt}}$ is the ground-truth mask. The

confidence scores are enforced to be strictly positive by setting $C_{i,j} \leftarrow 1 + \exp(C_{i,j})$. This enables a confidence estimate without explicit supervision.

We also predict binary object masks from novel viewpoints, and use a binary cross entropy loss to supervise it during training.

$$\mathcal{L}_{\text{mask}} = \sum_{i \in [0, W-1]} \sum_{j \in [0, H-1]} \left( -m_{i,j}^{\text{gt}} \log(m_{i,j}) - (1 - m_{i,j}^{\text{gt}}) \log(1 - m_{i,j}) \right) \qquad (4)$$

Here, $m_{i,j}$ is the predicted object mask after a sigmoid operation, and $m_{i,j}^{\text{gt}}$ is the ground-truth object mask. Finally, we combine the depth and mask losses with a sum weight $\lambda_{\text{mask}}$.

$$\mathcal{L}_{\text{total}} = \mathcal{L}_{\text{depth}} + \lambda_{\text{mask}} \mathcal{L}_{\text{mask}} \qquad (5)$$

### 4.3 View sampling

To construct a set of query views to sample, we fit a tight bounding box to the input view point map and sample points on a sphere with radius $\lambda_{bb} r_{bb}$ around the box's center. Here, $\lambda_{bb}$ is a tunable parameter, and $r_{bb}$ is half the length of the bounding-box diagonal. We found degenerate cases where the bounding box is too small, thus we clip the radius to be at least $\lambda_{\text{cam}} r_{\text{cam}}$, where $r_{\text{cam}}$ is the distance from the camera to the center of the bounding box and $\lambda_{\text{cam}}$ is a tunable hyperparameter. We sample points evenly on a cylindrical equal-area projection of the sphere to improve the coverage of the scene. We don't include the input point map in our predictions as it likely contains noise and artifacts. Instead, query RaySt3R with the input view and include it in our predictions.

### 4.4 Merging predictions

After predicting depth maps and object masks for all novel views, we merge them to produce a complete 3D shape. We merge the depth maps by accounting for occlusions, RaySt3R's predicted masks, and the confidence scores.

**Occlusion handling**: First, we filter the points in each novel view to only parts of the scene that were not visible in the input image (i.e., those points occluded by the input view's foreground mask $M^{\text{input}}$ and depth map $D^{\text{input}}$). Each point $q_{n,i,j}$ is defined as the point predicted by the $n$-th novel view at pixel $(i, j)$. Its projection in the input view is given by $p_{n,i,j} = K_{\text{input}} P_n^{\text{input}} h(q_{n,i,j})$, where $P_n^{\text{input}}$ transforms points from the $n$-th novel view to the input view. We define each entry of the mask as:

$$m_{n,i,j}^{\text{occ}} = \begin{cases} 1 & \text{if } (p_{n,i,j})_z > D_{i,j}^{\text{input}} \text{ and } M_{i,j}^{\text{input}} = 1 \\ 0 & \text{otherwise} \end{cases} . \qquad (6)$$

**RaySt3R predicted masks**: Even with the occlusion constraint, the object mask from a novel view is largely unknown. For example, any observed surface could be a thin plate or a rich 3D object. We use the predicted mask from RaySt3R to filter the points, by thresholding the predicted mask $m_{i,j}^{\text{RaySt3R}} \in [0, 1]$ at 0.5.

**Confidence scores**: RaySt3R's architecture enables unsupervised confidence scores for each pixel in the predicted depth maps. Confidence scores are typically used to reduce edge-bleeding in dense ViT predictions [48, 45], or to exclude out-of-distribution objects such as specularities. With the same objective, we threshold $c_{i,j}^{\text{RaySt3R}}$ at $\tau$ for all experiments, more analysis is provided in Section 5.8.

**Final mask**: The final valid mask for a given novel view is obtained by setting

$$m_{n,i,j} = m_{n,i,j}^{\text{occ}} \cdot \mathbf{1}\left[ m_{n,i,j}^{\text{RaySt3R}} > 0.5 \right] \cdot \mathbf{1}\left[ c_{n,i,j}^{\text{RaySt3R}} > \tau \right] \qquad (7)$$

We obtain our final 3D reconstruction by aggregating valid points across all novel views.

## 5 Results

### 5.1 Training dataset

RaySt3R's training procedure requires a large number of camera pairs to scale zero-shot to the real world. It requires an RGB image for the input view and depth maps, intrinsics, and extrinsics for

both cameras. We leverage existing synthetic datasets from FoundationPose [52] and OctMAE [18]. OctMAE [18] places GSO [8] and Objaverse [7] objects in synthetic scenes, and provide a single rendered image and depth map for each scene. FoundationPose has separate GSO and Objaverse splits, we only use the GSO split. For both datasets, we use the Objaverse and GSO meshes to render depth maps from novel views. Our dataset spans 251 k unique scenes, 12 k objects, and 11 M novel depth maps rendered for supervision.

## 5.2 Evaluation datasets

We evaluate RaySt3R on synthetic and real-world datasets. Following OctMAE [18], we evaluate on subsets of evaluation splits of the YCB-Video [56] (900 frames), HOPE [42] (50 frames), and HomebrewedDB [21] (1,000 frames) datasets. They are real-world 6D pose estimation datasets with noisy depth maps and imperfect masks, including common objects such as boxes and cylinders, as well as items of complex geometries such as metal parts. For results on synthetic data, we evaluate on evaluation split of the OctMAE [18] (1,000 frames) dataset test split. We notice edge artifacts in the masks introduced due to data compression in the original work [18].

## 5.3 Data augmentation

Synthetic training data lacks noise and other artifacts as present in the real world. We therefore apply data augmentation during training to better bridge the sim-to-real gap. Inspired by [52, 51, 6], we apply a set of augmentations to the input views at training time. For depth maps, we randomly apply Gaussian noise, add holes, and shift the pixel coordinates [2, 6]. For the RGB image, we randomly vary brightness and contrast, and apply a per-channel salt and pepper noise and Gaussian noise.

## 5.4 Implementation details

We train RaySt3R on $8\times$ 80-GB A100 GPUs for 18 epochs, totaling approximately 20 million scene iterations. We set the batch size to 10 per GPU, and a learning rate of $1.5 \times 10^{-4}$ with a half-cosine learning-rate schedule, starting with one warm-up epoch and using an AdamW optimizer [28]. We use a ViT-B model with patch size 16, embedding dimension 768, 12 heads, 12 cross-attention layers, but 4 self-attention layers to save on compute. We select the ViT-L with registers for DINOv2 [33].

We set $\lambda_{bb} = 1.3$ and $\lambda_{cam} = 0.7$ for all real-world datasets, and $\lambda_{bb} = 2.5$ and $\lambda_{cam} = 1.2$ for the OctMAE dataset. The parameters are chosen to be larger for the OctMAE dataset, as the input view is typically placed very close to the objects with severe occlusions. We set the confidence threshold $\tau = 5$ for all experiments, and sample 22 views in total. During training we set the confidence parameter $\alpha = 0.2$, $\lambda_{mask} = 0.1$. Inference takes less than 1.2 seconds on a single RTX 4090 GPU, and can be further reduced by querying fewer views.

## 5.5 Baselines

We compare RaySt3R against the state-of-the-art in 3D shape completion. OctMAE introduced a novel 3D MAE algorithm, and also trained prior shape completion models on their novel dataset. We compare against OctMAE, and the prior works they trained, i.e., VoxFormer [26], ShapeFormer [57], MCC [53], ConvONet [35], POCO [3], AICNet [24], Minkowski [5], and OCNN [46].

We also compare against SceneComplete [1], which uses a combination of foundation models to produce complete geometry. The authors leverage a VLM and Grounded-SAM [37] to produce object-level masks, image inpainting to fill in occluded regions, an image-to-3D model to produce a 3D mesh, and finally FoundationPose for 3D alignment.

We also benchmark against Unique3D [54] and TRELLIS [55], which are recent image to 3D models. Finally, we compare against 'Layered Ray Intersections' (LaRI) [25], which introduced the concept of layered point maps to predict multiple points on each camera ray. Unique3D, Trellis, and Lari predict points in canonical coordinates, we align the predictions with the ground truth 3D using first a brute-force search for a similarity transform, followed by ICP [40]. Note that we do not perform such a registration for RaySt3R, but provide this to baselines to give them the benefit of the doubt. LaRI [25], Unique3D [54] and TRELLIS [55] were not trained on cluttered scenes.

While LaRi [25] and Unique3D [54] do not require foreground masks for reconstructing objects, we observe that TRELLIS [55] tends to reconstruct the entire scene. Therefore, we compare TRELLIS with a masked image input and a raw image input. We also attempted to evaluate ViewCrafter [59] on this task, but the images produced by the video diffusion model were of too poor quality to perform the evaluation. We provide more details in the supplementary material.

## 5.6 Quantitative results

Following prior work [18], we evaluate the zero-shot generalization performance of all methods using chamfer distance (CD) and F1-Score@10mm (F1). Detailed formulations of the metrics are provided in the supplemental material. We present the quantitative results of this evaluation in Table 1. RaySt3R consistently outperforms all baselines across both synthetic and real-world scenes. The strongest baseline, OctMAE [18], performs competitively, however RaySt3R surpasses it across all metrics, by 20 % to 44 % in CD. SceneComplete [1] proves to be a fragile pipeline, therefore not yielding competitive results. We were unable to produce SceneComplete results on HOPE, as the pipeline requires an intractable amount of VRAM for cluttered scenes. LaRI [25], Unique3D [54], and TRELLIS [55] show better performance than SceneComplete [1], but also do not produce competitive results. Feeding masked RGB images to TRELLIS [55] outperforms raw image inputs on all datasets except HomebrewedDB [21]. We show common failure modes in Section 5.7.

We also compute the standard deviation of the chamfer distance across all real-world datasets for each method and observe that our model exhibits the lowest standard deviation (1.74 mm), followed by OctMAE [18] (2.38 mm) and Unique3D [54] (8.11 mm).

Table 1: Quantitative evaluation of multi-object scene completion on synthetic and real-world datasets. We evaluate on the test split of OctMAE [18], and the BoP benchmarks YCB-Video [56], HOPE [42], and HomebrewedDB [21]. We report chamfer distance (CD) [mm] and F1-Score@10mm (F1). The first section contains numbers copied from OctMAE [18], the second section contains recent works we evaluated. For alignment of LaRI [25], Unique3D [54], and TRELLIS [55] with the ground truth mesh, we apply brute force search followed by ICP [40]. We evaluate TRELLIS [55] with masked and unmasked RGB inputs. SceneComplete [1] runs out of VRAM on HOPE [42]. The results suggest RaySt3R outperforms all baselines.

| | Synthetic | | Real | | | | | |
| | OctMAE [18] | | YCB-Video [56] | | HB [21] | | HOPE [42] | |
| Method | CD↓ | F1↑ | CD↓ | F1↑ | CD↓ | F1↑ | CD↓ | F1↑ |
|---|---|---|---|---|---|---|---|---|
| VoxFormer [26] | 44.54 | 0.382 | 30.32 | 0.438 | 34.84 | 0.366 | 47.75 | 0.323 |
| ShapeFormer [57] | 39.50 | 0.401 | 38.21 | 0.385 | 40.93 | 0.328 | 39.54 | 0.306 |
| MCC [53] | 43.37 | 0.459 | 35.85 | 0.289 | 19.59 | 0.371 | 17.53 | 0.357 |
| ConvONet [35] | 23.68 | 0.541 | 32.87 | 0.458 | 26.71 | 0.504 | 20.95 | 0.581 |
| POCO [3] | 21.11 | 0.634 | 15.45 | 0.587 | 13.17 | 0.624 | 13.20 | 0.602 |
| AICNet [24] | 15.64 | 0.573 | 12.26 | 0.545 | 11.87 | 0.557 | 11.40 | 0.564 |
| Minkowski [5] | 11.47 | 0.746 | 8.04 | 0.761 | 8.81 | 0.728 | 8.56 | 0.734 |
| OCNN [46] | 9.05 | 0.782 | 7.10 | 0.778 | 7.02 | 0.792 | 8.05 | 0.742 |
| OctMAE [18] | 6.48 | 0.839 | 6.40 | 0.800 | 6.14 | 0.819 | 6.97 | 0.803 |
| LaRI [25] | 39.22 | 0.283 | 11.41 | 0.658 | 22.23 | 0.414 | 18.64 | 0.528 |
| Unique3D [54] | 44.62 | 0.244 | 17.56 | 0.468 | 25.41 | 0.329 | 26.37 | 0.322 |
| TRELLIS (w/ mask) [55] | 65.43 | 0.224 | 22.44 | 0.454 | 36.12 | 0.364 | 19.46 | 0.470 |
| TRELLIS (w/o mask) [55] | 69.45 | 0.221 | 31.45 | 0.345 | 29.98 | 0.360 | 20.87 | 0.438 |
| SceneComplete [1] | 81.57 | 0.289 | 96.63 | 0.359 | 85.81 | 0.416 | N/A | N/A |
| RaySt3R (ours) | **5.21** | **0.893** | **3.56** | **0.930** | **4.75** | **0.889** | **3.92** | **0.926** |

## 5.7 Qualitative results

Figure 3 shows qualitative results of RaySt3R on real-world datasets. The results suggest that RaySt3R is capable of generating high-quality 3D predictions, generating more consistent and complete results compared to the baselines. The most competitive baseline, OctMAE [18], produces viable but oversmoothed object shapes due to its low-resolution 3D MAE grid. Furthermore, TRELLIS [55], Unique3D [54] and LaRI [25] struggle with relative object placement and aspect ratios. They also fail for certain out-of-distribution objects, and occasionally fail to register to the ground truth point cloud.

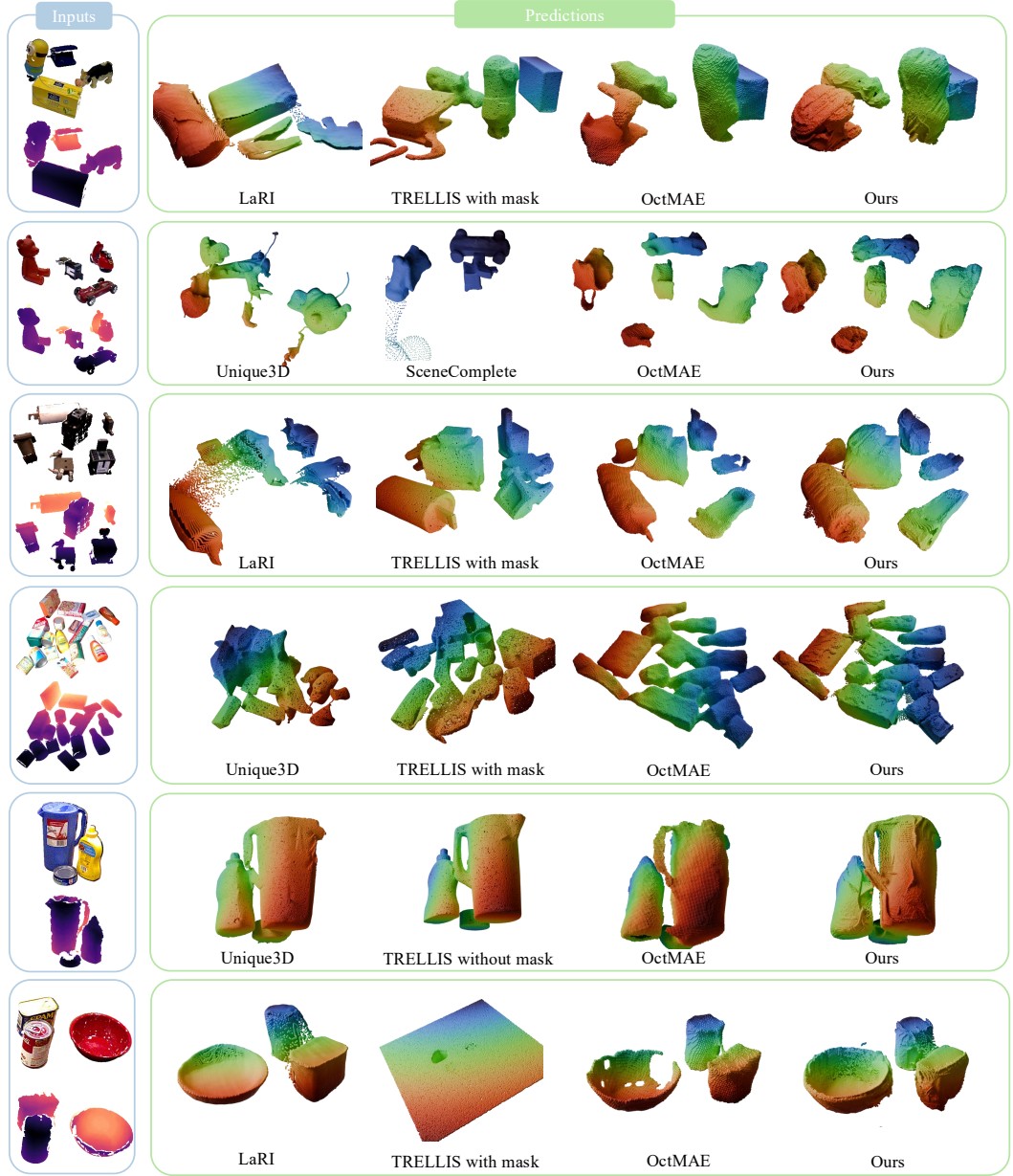

Figure 3: Qualitative results of RaySt3R in real-world multi-object scenes. For each scene, we select a subset of methods to preserve visual clarity, but we share all method predictions in the supplemental material. The results suggest RaySt3R produces the most geometrically accurate shapes compared to the baselines. The most competitive baseline, OctMAE [18], tends to predict softer shapes as a result of its coarse 3D MAE grid. We observe TRELLIS [55], Unique3D [54], and LaRI [25] struggle with relative object placing, aspect ratios, and out-of-distribution objects, occasionally leading to incorrect registration to the ground truth points. Finally, SceneComplete [1] proves to be brittle to single points of failure such as missing object masks.

Interestingly, TRELLIS [55] may predict table surfaces even for masked input images with no visible table. SceneComplete [1] proves to be brittle to single-point failures such as missing object masks.

## 5.8 Ablation studies

**Training ablations**: Table 2 summarizes our training ablation results, with all models trained under the same setup for roughly 20 million scene iterations. Training a ViT-S model (384-dim, 6 heads) leads to a performance drop. Disabling data augmentation further degrades zero-shot

Table 2: Ablation study on RaySt3R training on the YCB-Video dataset [56]. The results suggest data scale, data diversity, DINOv2 [33] features, data augmentation, and model size affect RaySt3R's performance.

| Method name | CD ↓ | F1 ↑ |
|---|---|---|
| RaySt3R (proposed) | **3.56** | **0.930** |
| ViT-S | 3.70 | 0.920 |
| No data augmentation | 3.89 | 0.916 |
| Train on 100k scenes | 4.30 | 0.894 |
| w/o DINOv2 [33] | 4.81 | 0.877 |
| Train on 226k GSO scene | 5.34 | 0.864 |

Table 3: Ablation study on RaySt3R view merging on the Oct-MAE test dataset [18]. We ablate querying the input view (Section 4.3), occlusion masking with the input mask, and finally RaySt3R's predicted masks. The results suggest all steps contribute to performance, especially the predicted masks.

| Query Input | Occ. Mask | Pred. Mask | CD↓ | F1↑ |
|---|---|---|---|---|
| ✓ | ✓ | ✓ | **5.21** | **0.893** |
| ✗ | ✓ | ✓ | 7.55 | 0.836 |
| ✓ | ✗ | ✓ | 7.69 | 0.855 |
| ✓ | ✓ | ✗ | 10.12 | 0.825 |
| ✗ | ✗ | ✗ | 73.17 | 0.641 |

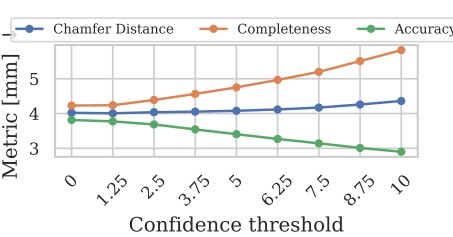

Figure 4: Confidence threshold vs error metrics averaged over all real-world datasets. The results suggest increasing the confidence threshold improves accuracy and degrades completeness.

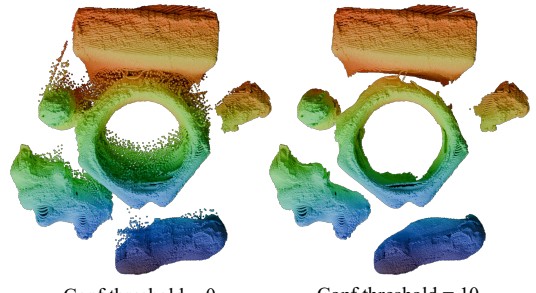

Conf threshold = 0          Conf threshold = 10

Figure 5: The impact of the confidence threshold on the predicted 3D points. This experiment suggests increasing the confidence threshold can aid in reducing the edge bleeding issue.

generalization to real-world data, highlighting its importance. We also compare training on 100k uniformly sampled scenes versus the full 226k GSO set. The smaller but more diverse 100k set performs better, emphasizing the value of data diversity. Finally, removing DINOv2 [33] inputs causes an additional decline, underscoring the benefit of pretrained features. We also train a model exclusively on the OctMAE split of our dataset, and find it achieves a CD of 5.87, and an F1 of 0.893 on the OctMAE test set. This result suggests RaySt3R outperforms the baselines even when trained on the same dataset.

**View merging ablations**: Table 3 shows the ablations on our view merging formulation. We ablate querying the input view (Section 4.3), using the input mask to detect occluded regions, and finally the use of RaySt3R's predicted mask. The results suggest that each component of our formulation contributes to shape completion performance, especially the predicted masks.

**Confidence** RaySt3R predicts a per-pixel confidence value, which can be used to filter the predictions. To understand the impact of the confidence threshold, we report chamfer distance, Completeness, and Accuracy for a range of thresholds, as depicted in Figure 4. The results suggest that confidence is a good proxy for error and that confidence can be effectively used to trade off accuracy and completeness. Depending on the application, the threshold can therefore be tuned accordingly, as some applications may be less tolerant to outliers requiring high accuracy, while others may require more complete predictions. For all prior experiments, we set confidence threshold $\tau$ to 5 for a balance between accuracy and completeness. Figure 5 shows a visualization of the impact of changing the confidence threshold on the predicted 3D points.

## 6  Conclusion and future work

We present RaySt3R, a novel approach to 3D shape completion from a single RGB-D image and foreground mask. RaySt3R learns to predict depth maps, object masks, and per-pixel confidence scores for novel viewpoints, which are fused to reconstruct complete 3D shapes. We benchmark RaySt3R on real-world and synthetic datasets, and compare it to the state-of-the-art in volumetric

and view-based methods. The results suggest that RaySt3R is capable of generating high-quality 3D predictions, outperforming the baselines across the board.

While the results suggest our view-based approach generates high-quality reconstructions, it also suffers from the common edge bleeding issue, adding noise to predictions. Our formulation also requires inferring high-quality novel view poses, which is a non-trivial task beyond objects on table tops. An advantage of RaySt3R's view-based approach is that it enables training on real-world data without the need for ground truth meshes. Training on real-world data may help generalize to more objects and adapt to real-world noise. Future work may also explore scaling up compute by exploring other architectures like diffusion transformers.

## 7 Acknowledgements

This work was generously supported by the Center for Machine Learning and Health (CMLH) at CMU, the NVIDIA Academic Grant Program, and the Pittsburgh Supercomputing Center. The authors would like to thank Mandi Zhao, Shun Iwase, Balázs Gyenes, Gerhard Neumann, Sergey Zakharov, Nikhil Varma Keetha, Jeff Tan and all members of the Momentum Robotics lab at CMU for providing useful feedback.

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
