# OpenReview forum: "RaySt3R: Predicting Novel Depth Maps for Zero-Shot Object Completion"
_NeurIPS.cc/2025/Conference — NeurIPS 2025 poster_

### Official Review · Reviewer_FXY9 · 2025-06-27

**Clarity:** 3
**Significance:** 2
**Originality:** 2
**Rating:** 4
**Confidence:** 4

**Summary:**

This paper proposes a view-based architecture followed by view-based sampling and point filtering to perform objection completion from a single RGB-D image.  It also proposes a new simulated multi-object dataset to help improve performance.

**Questions:**

The key topics that I would like clarifaction on are:

1.	OctMAE has slightly worse performance across metrics while being trained on significantly less data (25k scenes and 1M images) compared to the proposed work (1.1M scenes and 12M images).  Since the two main contributions of the paper are the RaySt3R architecture and the dataset, these should be decoupled to understand how much performance is gained from each individually.  Ideally, OctMAE would be trained on this new dataset and RaySt3R would be trained on the OctMAE dataset so that this is clear.

2.	The Chamfer distance metric raises several questions.  In Figure 4, the optimal Chamfer distance comes at a confidence threshold of 0.  However, this was not the threshold that was used, and instead a confidence threshold of 5 is used despite giving worse performance.  Visually, Figure 5 shows that a confidence threshold of 0 gives very noisy results with many points far from the surface.  This raises the question that the metrics used are not indicative of good results.  One potential issue is that L1-Chamfer is far too lenient with outliers, and L2 and RMSE should be reported instead.  Furthermore, completeness and accuracy are only shown in Figure 4, but this could be reported in Table 1 as there is a trade-off between the two as shown in the plot.

3.	More discussion on limitations of view-based methods compared to 3D representations would benefit the paper.  The limitations section at the end of the conclusion does not dive into the method, but is rather a generic statement on the need for real-data and scaling that could apply to any machine learning method.  Using a view-based method is an interesting data point, but the benefits and drawbacks could be discussed in much greater detail.  For example, the abstract says the approach addresses 3D consistency that other methods lack, but it is unclear why a ray-based approach, which requires additional occlusion filtering here, is more consistent than a volumetric one.  The first clip in the video also demonstrates that  views cannot recover the bottom of the objects which are left empty.

**Ethical Concerns:**

["NO or VERY MINOR ethics concerns only"]

**Final Justification:**

I appreciate that the authors covered many points of discussion during the rebuttal, specifically:

 - Training RaySt3R on the OctMAE dataset for fair comparison
 - Using a unified set of parameters across all datasets
 - Clarifications of the dataset compared to the one proposed in OctMAE
 - Agreeing to give discussion on view-based vs. volumetric representations for scene completion
 - Inclusion of a limitations section beyond the generic ones listed in the conclusion
 - Additional details on metrics

I have upgraded my score to borderline accept, as overall the paper provides valuable contributions.

The main factor for not raising the score beyond borderline accept is that it is still unclear why a confidence threshold that gives worse average results across all datasets is selected (Table 6 in the response to my review).  This somewhat diminishes the Chamfer distance as a reliable metric. It is used to signal how RaySt3R outperforms baselines, but then RaySt3R is designed to not fully optimize this metric.

**Limitations:**

As mentioned above, more discussion on limitations of view-based methods would benefit the paper.

**Quality:**

2

**Strengths And Weaknesses:**

Overall, the paper is well-written and the method is straightforward to understand.  The visualizations are also clear and visually appealing.

I will discuss some questions and clarifications that I have below:

•	In general, there should be more discussion on the drawbacks of view-based approaches and how they differ compared to 3D representations.  The proposed method claims to address 3D consistency, but this is a potential problem with view-based methods compared to 3D representations, such as voxel grids.  Furthermore, view-based methods have the problem that the views must be selected to ensure coverage, but in some cases, this may be extremely challenging.  Even in the video, the bottom of objects and part of the pitcher are missing.  In very cluttered scenes, views have to be selected such that the rays pass between two objects, which with contact may be impossible.

•	While it is mentioned that RaySt3R does not need volumetric supervision, it still requires complete geometry,  It does not seem like a significant drawback that volumetric methods need to precalculate from the complete geometry since this is at training time and has no effect at test time.

•	As the paper claims the approach is efficient, a runtime comparison with others would be helpful.  View-based methods will also have complexity linear in the number of target views, which may need to vary for different scenes depending on the amount of clutter.  22 views in 1.2 seconds is fairly efficient, but comparing to other approaches would strengthen this.

•	How does the random sampling guarantee coverage? Some results in the video for example show insufficient coverage.

•	It is mentioned that the input point map is not included in the predictions.  Does this mean there is noise in the depth maps or do the predictions not align with the input point map?  The occlusion filter is also used to remove points that would be visible in the original view, but is this due to some predicted points not being 3D consistent due to the view-based method?

•	It is difficult to judge how OctMAE compares with the proposed method, as they are trained on very different amounts of data with different augmentations.  RaySt3r uses 1.1 million scenes and 12 million views, while OctMAE is trained on 25k scenes and 1M images.  Since OctMAE is the closest in terms of performance, evaluating on equal footing with respect to training data would demonstrate if the data, the architecture, or both are the key to improved performance.

•	Sampling parameters are varied between real and OctMAE datasets.  Do other methods use dataset specific parameters? If not, a unified set of parameters should be used for fairness.

•	L1-Chamfer is used in Figure 4.  Is this the Chamfer distance used for all experiments? In Figure 4, the best Chamfer distance is with a confidence threshold of 0, but a worse performing confidence threshold of 5 is used for experiments instead.  In Figure 5, a 0 threshold gives inconsistent and noisy results.  This leads to the question whether the metrics used are indicative of good 3D completion, as the best performing threshold leads to worse visual results.  L1-Chamfer is likely too forgiving for outliers and noisy points, so L2 and RMSE should potentially be used instead.

---

> ### Author Rebuttal · Authors · 2025-07-31
>
> Thank you FXY9 for these great questions, we will modify the paper to reflect our answers below:
>
> Q1. **Comparison against OctMAE**:  	Following your suggestion, we trained RaySt3R only on the OctMAE dataset and with only the OctMAE augmentations. The table below shows RaySt3R trained on either dataset outperforms OctMAE (**bold** is best, _italics_ is second best).
>
> | Method             | Trained on | CD $\downarrow$ | F1 $\uparrow$ |
> | ------------------ | ---------- | --------:| -----------:|
> | **OctMAE**         | OctMAE     | 6.48     | 0.839       |
> | **Ours (RaySt3R)** | OctMAE (new) | _5.87_   |  _0.877_    |
> | **Ours (RaySt3R)** | RaySt3R    | **5.21** | **0.893**   |
> _Table 1. Model and training dataset comparision_
>
> While we did not have the resources to train OctMAE on our dataset, we believe that this experiment nevertheless shows the contribution of our method.
>
> In retrospect, our description of the datasets may have been unclear.
> Our dataset uses the exact same objects as OctMAE, and only 10\% more input images.  "1.1 million scenes", refers to the unique RGB-D input images; but the actual scenes are only 251k.  Also, "12 M views" includes both the input images and the depth maps used for supervision.  See table below.
>
> |  | **OctMAE** dataset | **RaySt3R** dataset |
> |---|---:|---:|
> | # RGB-D Input Images | 1 M | 1.1 M |
> | # Objects |   12 k   |    12 k  |
> | # Scenes (object configurations) | 25 k | 251 k |
> | Supervision | Sampled mesh | 11 M depth maps |
> _Table 2. Training dataset size comparision_
>
> Q2. **Evaluation Metrics**: We follow prior work by computing the Chamfer distance with the L2 norm (the Euclidean distance) for both accuracy and completeness.
> See Equations (1)-(3) in the supplemental material.
> (In the paper, the reference to "Chamfer-L1" is an error which we will correct in the final version of the paper.)
>
> As requested, to further study the impact of the introduced noise, the tables below provide the CD computed based on RMSE, as well as the accuracy and completeness.
> These additional metrics show that RaySt3R consistently outperforms the baselines.
>
>
> | Method            | OctMAE | YCB-Video | HomebrewedDB | HOPE  |
> | ----------------- | ------:| ---------:| ------------:| -----:|
> | Unique3D          |  62.23 |     24.16 |        34.27 | 36.12 |
> | TRELLIS with mask |  84.94 |     31.55 |        45.55 | 27.34 |
> | TRELLIS           |  88.32 |     42.26 |        40.32 | 30.53 |
> | LaRI              |  55.88 |     15.72 |        30.54 | 25.26 |
> | OctMAE            |  21.72 |     11.31 |        10.30 | 14.52 |
> | Ours (RaySt3R)           |   **9.33** |      **6.17** |         **7.85** |  **6.37** |
> _Table 3. RMSE results_
>
> | Method            | OctMAE | YCB-Video | HomebrewedDB | HOPE  |
> | ----------------- | ------:| ---------:| ------------:| -----:|
> | Unique3D          |  47.29 |     20.27 |        27.72 | 30.18 |
> | TRELLIS with mask |  48.12 |     22.43 |        26.22 | 20.74 |
> | TRELLIS           |  45.34 |     28.20 |        23.23 | 20.28 |
> | LaRI              |  41.21 |     11.74 |        24.60 | 19.66 |
> | OctMAE            |  20.27 |      8.49 |         8.69 |  7.69 |
> | Ours (RaySt3R)           |   **7.80** |      **3.67** |         **5.93** |  **4.66** |
> _Table 4. Completeness results_
>
>
>
> | Method            | OctMAE | YCB-Video | HomebrewedDB | HOPE  |
> | ----------------- | ------:| ---------:| ------------:| -----:|
> | Unique3D          |  45.87 |     14.86 |        23.09 | 22.57 |
> | TRELLIS with mask |  84.11 |     23.46 |        42.71 | 19.77 |
> | TRELLIS           |  94.42 |     35.27 |        38.19 | 23.82 |
> | LaRI              |  42.61 |     11.08 |        19.87 | 17.62 |
> | OctMAE            |   5.93 |      5.06 |         4.28 |  7.21 |
> | Ours (RaySt3R)           |   **2.63** |      **3.45** |         **3.58** |  **3.19** |
> _Table 5. Accuracy results_
>
>  **Confidence threshold:**
> Fig 4 shows a tradeoff: a lower confidence threshold yields better Chamfer distance but worse accuracy. To avoid unfairly choosing hyperparameters from the test data, we arbitrarily selected a threshold of 5 for all experiments, qualitative and quantiative. The table below shows the Chamfer distance for each scene with different confidence thresholds. Overall, RaySt3R is not sensitive to the choice of confidence threshold and consistently outperforms the baselines under all confidence thresholds.
>
> | Method | Conf | YCB-Video | HomebrewedDB | HOPE |
> |--------|------:|------------:|--------------:|------:|
> | Ours (RaySt3R) | 0.00 | 3.59 | **4.49** | 3.99 |
> | Ours (RaySt3R) | 1.25 | **3.53** | 4.50 | 3.99 |
> | Ours (RaySt3R) | 2.50 | 3.55 | 4.58 | 3.98 |
> | Ours (RaySt3R) | 3.75 | 3.56 | 4.67 | 3.94 |
> | Ours (RaySt3R) | 5.00 | 3.56 | 4.75 | **3.92** |
> | Ours (RaySt3R) | 6.25 | 3.57 | 4.85 | 3.94 |
> | Ours (RaySt3R) | 7.50 | 3.58 | 4.95 | 3.97 |
> | Ours (RaySt3R) | 8.75 | 3.62 | 5.11 | 4.05 |
> | Ours (RaySt3R) | 10.00 | 3.65 | 5.27 | 4.16 |
> | OctMAE  | N/A  | 6.40 | 6.14 | 6.97 |
> _Table 6. Confidence ablation_
>
> Q3. **View-based vs volumetric approachs**: We agree the pros and cons of view-based completion vs 3D representations should be emphasized more. The advantages of view-based completion are: (1) the ability to query an arbitrary number of views to trade off quality and inference speed; (2) the ability to supervise directly on real-world data without GT meshes; and (3) pixel-aligned and more accurate prediction compared to volumetric methods. In terms of limitations, while view-based methods may suffer from multi-view consistency issue such as edge bleeding, our proposed  confidence-based thresholding and multi-view fusion strategies effectively enhance the raw prediction from the network. Reconstruction results evaluated under multiple metrics indicate our method consistently outperforms volumetric approaches such as OctMAE. In the original abstract, the limited multi-view consistency refers to the other view-based methods such as ViewCrafter, which has been compared against and outperformed by our method in the supplemental material. However, we do admit there are a few other limitations which we will discuss more in the updated version of the paper, including: (1) deciding where to sample novel views involves more designs than volumetric methods; (2) when a relatively large number of views are queried, it leads to slower inference times; (2) generalizing to objects far beyond the current training dataset, such as transparent
> objects.
>
> **Weaknesses**
> We respond to other points raised below:
> * **Video animation**: We added the first animation to the video only for illustrative purposes, showing only 3 queried views instead of 22. To see the full reconstruction, please skip to 2:21 in the video or Figure 15 in the supplemental material. RaySt3R successfully reconstructs the bottom of the object because it samples from the viewing sphere (even though the bottom is not shown).
> * **Training without complete geometry**: While we train RaySt3R on a synthetic dataset, our method (unlike volumetric methods) does not require ground-truth meshes and can be trained on 2 or more depth images of a scene (as would be available from real-world depth cameras). We will clarify this in the final version of the paper.
> * **Random sampling**: We worded this incorrectly.  We did not use random sampling, since we found it is more likely to miss areas of the scene and introduce non-determinism.  We place query views on a grid evenly spaced on a cylindrical equal-area projection of a sphere.
> * **Noisy input depth**: We do not use the 3D points from the input depth map. In the real datasets they tend to be noisy, and introduce floaters especially at edges of objects. There are no 3D consistency issues. We conduct an experiment by appending the input depth map to RaySt3R's predictions in the YCB-V dataset, and observe the Chamfer distance increases from 3.56 [mm] to 3.66 [mm].
> * **Runtime**: We appreciate your suggestion to provide a more comprehensive runtime analysis. In the table below, we provide additional information as benchmarked on our own hardware. SceneComplete is more challenging to benchmark, as it relies on an OpenAI API call to use their VLM. We find RaySt3R with flash attention runs faster than vanilla torch attention (568 ms vs 1200 ms).
>
>
> | Method            |  Runtime [ms]      |
> | ----------------- | ------------------:|
> | OctMAE            |        56            |
> | LaRI              |       283            |
> | TRELLIS           |  11,409               |
> | SceneComplete* |              $\sim$ 30,000     |
> | Ours (RaySt3R) w Torch Atnn |              1200 |
> | Ours (RaySt3R) w Flash Atnn |              568 |
> _Table 7. Runtime comparison on 1 $\times$ RTX 4090. *Per object, as reported by original paper._
>
> - **Unified set of parameters**: Not all code for the listed baselines is available, hence we are unsure whether hyperparameters were varied for each dataset. Nevertheless, we study RaySt3R's performance with a single set of hyperparameters instead of two sets for synthetic and real datasets. This experiment (see table below) suggests RaySt3R outperforms the baselines even with a single set of hyperparameters.
>
> | Method                          | OctMAE | YCB-Video | HomebrewedDB | HOPE |
> | ------------------------------- | ------:| ---------:| ------------:| ----:|
> | OctMAE                          |   6.48 |      6.40 |         6.14 | 6.97 |
> |   RaySt3R                       |   **5.21** |      **3.56** |        **4.75**  | **3.92** |
> | RaySt3R w single parameter set  |   **5.21** |     4.19  |         5.20     |  4.66    |
> _Table 8. Hyper parameter study, Chamfer Distance [mm]_

---

> > ### Comment · Reviewer_FXY9 · 2025-08-04
> >
> > Thank you for the detailed response.  I believe that the paper has been strengthened by the comments by the authors regarding:
> > - RaySt3R being trained on the OctMAE dataset
> > - Clarification of the dataset
> > - Additional metrics (accuracy, completeness, RMSE)
> > - Discussion on view-based vs. volumetric/3D representations
> > - Timing breakdown of different methods
> >
> > Training OctMAE on the proposed dataset would help complete the story of why the proposed dataset is important, but it is understandable that this is not possible in the duration of the rebuttal.
> >
> > One additional question I have relates to the supervision.  As mentioned, RaySt3R does not need complete geometry, and could be trained using just two depth images of a scene.  The paper places great emphasis on being “zero-shot”, but if the method can be trained on real images, why is there a focus on only using simulated data and being able to generalize to real-data?  This also raises the question on whether the proposed simulated dataset leads to improved performance over existing real datasets.
> >
> > I also agree with the question by bxUS on whether this method should be emphasized as zero-shot instead of sim-to-real for example.  The task of predicting depth maps/object masks given a single RGB-D is the same at training time as it is as test time.
> >
> > Regarding the confidence thresholds and metric, the additional table is useful to see.  However, it is interesting to see that having no threshold is always better than having a threshold of 10, which is shown in Figure 5.  In one dataset, no threshold is also the best performing, and the best overall average across datasets is a confidence threshold of 1.25 (4.01) followed by 0.0 (4.02), while 10.0 (4.36) is the worst overall average. Looking at Figure 5 though, it is clear that having a threshold of 10 is preferred to the 0 threshold.  I think this still raises the question of whether the metrics used are actually reflecting what we want in this task.  If having a noisy point cloud with edge bleeding is nearly the best performing method in terms of metrics, but produces clearly visually poor results, are the metrics measuring what we really want?
> >
> > While RaySt3R performs better than OctMAE across thresholds, the left side of Figure 5 looks visibly worse than OctMAE results despite having better Chamfer distance.  The RMSE and L2 results show that RaySt3R outperforms in those as well, but maybe these metrics also do not tell the full story.  Metrics such as the percentage of points below some error threshold (as in depth estimation) with varying thresholds may be helpful but in general I am interested to see if there is a metric that can reconcile the visualizations with the quantitative results.

---

> ### Author Response · Authors · 2025-08-05
>
> Thank you for responding to our rebuttal. We answer your questions below:
>
> - **How to explain the visualizations and metrics?** We visualize the results using a colored point cloud, where even a small number of outliers can make the point cloud appear noisy in visualizations. Although this small number of noisy predictions may not have a large effect on the metrics, OctMAE tends to predict overly smooth shapes and even produces holes (e.g., the last row of Figure 3). Overly smooth or partially missing geometry will have a large impact on all metrics considered. These inaccurate predictions are less noticeable on the static figure, but may cause problems in downstream applications.
> - **How to use the confidence threshold?** While previous works such as VGGT and DUSt3R leverage confidence similarly, we are (to the best of our knowledge) the first to study its implications. The confidence threshold enables users to strike a balance between accuracy and completeness. With Figures 4 and 5, as well as the qualitative Figures, we aim to give the reader insight into this tradeoff. Downstream applications relying on visual quality may choose a higher confidence threshold, while 3D scene completion in robotics may prioritize completeness with a lower threshold. Thanks to the experiments you suggested, our results show RaySt3R outperforms the baselines in both accuracy and completeness with a single confidence threshold. The best confidence threshold for the average Chamfer distance (accuracy and completion) depends on the scene and dataset, as the network's absolute confidence values vary. Investigating the confidence threshold even further, or proposing alternative confidence formulations, would be an interesting topic for future research.
> -   **Are there better metrics to use?** We follow the evaluation protocol from prior works, such as OctMAE, under commonly considered metrics. By your suggestion, we compute the F1 scores at various thresholds and report them in the tables below. RaySt3R outperforms all baselines consistently under varying thresholds. This confirms that edge bleeding noisy points only form a small fraction of the total prediction, while accurate surface prediction occupied by the majority of points is more critical for the metrics and downstream applications. In practice, we envision that those minor outlier points are relatively easy to remove if desired, by means such as TSDF or statistical outlier removal. Please let us know if there are other metrics you would like us to evaluate.
>
> | Method | OctMAE | YCB-Video | HomebrewedDB | HOPE |
> | --- | --- | --- | --- | --- |
> | Unique3D | 0.13 | 0.27 | 0.18 | 0.17 |
> | TRELLIS with mask | 0.13 | 0.28 | 0.21 | 0.25 |
> | TRELLIS | 0.12 | 0.20 | 0.21 | 0.23 |
> | LaRI | 0.16 | 0.42 | 0.24 | 0.30 |
> | OctMAE | 0.74 | 0.64 | 0.66 | 0.64 |
> | RaySt3R | **0.76** | **0.81** | **0.75** | **0.78** |
> _Table 1. F1@5mm $\uparrow$_
>
> | Method | OctMAE | YCB-Video | HomebrewedDB | HOPE |
> | --- | --- | --- | --- | --- |
> | Unique3D | 0.24 | 0.47 | 0.33 | 0.32 |
> | TRELLIS with mask | 0.22 | 0.44 | 0.36 | 0.44 |
> | TRELLIS | 0.21 | 0.34 | 0.35 | 0.42 |
> | LaRI | 0.27 | 0.66 | 0.41 | 0.53 |
> | OctMAE | 0.85 | 0.79 | 0.81 | 0.79 |
> | RaySt3R | **0.89** | **0.93** | **0.89** | **0.93** |
> _Table 2. F1@10mm $\uparrow$_
>
> | Method | OctMAE | YCB-Video | HomebrewedDB | HOPE |
> | --- | --- | --- | --- | --- |
> | Unique3D | 0.39 | 0.70 | 0.54 | 0.54 |
> | TRELLIS with mask | 0.34 | 0.63 | 0.54 | 0.67 |
> | TRELLIS | 0.34 | 0.52 | 0.53 | 0.65 |
> | LaRI | 0.43 | 0.84 | 0.62 | 0.73 |
> | OctMAE | 0.92 | 0.92 | 0.93 | 0.92 |
> | RaySt3R | **0.95** | **0.98** | **0.96** | **0.98** |
> _Table 3. F1@20mm $\uparrow$_
>
> | Method | OctMAE | YCB-Video | HomebrewedDB | HOPE |
> | --- | --- | --- | --- | --- |
> | Unique3D | 0.50 | 0.82 | 0.68 | 0.68 |
> | TRELLIS with mask | 0.44 | 0.74 | 0.66 | 0.78 |
> | TRELLIS | 0.43 | 0.64 | 0.66 | 0.77 |
> | LaRI | 0.54 | 0.91 | 0.74 | 0.82 |
> | OctMAE | 0.95 | 0.97 | 0.96 | 0.96 |
> | RaySt3R | **0.97** | **0.99** | **0.98** | **0.99** |
> _Table 4. F1@30mm $\uparrow$_
>
> - **Any other methods to reduce noise?** The noise we observe with RaySt3R is similar to that produced by a depth camera. In the open-source release, we will include an off-the-shelf TSDF depth map merging algorithm, which will output smoother meshes for downstream applications.
> - **Why focus on zero-shot performance?** We agree we have over-emphasized the zero-shot aspect, and commit to toning down the claims. We meant to highlight encouraging performance on the real-world datasets with real-world sensor noise and unseen objects. We believe this is an encouraging result, which may improve further when combining synthetic and real-world training data.

---

> > ### Author Response · Authors · 2025-08-06
> >
> > Thank you for your engagement in the rebuttal thus far. As the author-reviewer discussions have been extended to August 8th, please let us know if you have any more questions that will help inform your decision.

---

### Official Review · Reviewer_G7Ap · 2025-06-28

**Clarity:** 3
**Significance:** 2
**Originality:** 2
**Rating:** 4
**Confidence:** 4

**Summary:**

This paper proposes a supervised learning method for predicting a full 3D model of an object from a single masked RGB-D image (the so-called shape completion problem). It does so by learning to predict new depth maps of a scene from different viewpoints, and combining multiple such views into the final 3D model.

**Questions:**

I think it would be important to clarify what new insights (besides its results) are to be gained from the paper. Are its methodological aspects, for example, likely to be usesful in other tasks, and why?

Please clarify the "zero-shot" aspects of the presentation since this paper has nothing to do with scene categories, and is just about reconstruction.

You appear to predict a point cloud, but the illustrations seem to be mesh renderings. Please clarify.

**Ethical Concerns:**

["NO or VERY MINOR ethics concerns only"]

**Final Justification:**

I remain unconvinced by the importance of the problem addressed in this paper, but the proposed solution for this type of problems is reasonable, and the authors have done a good job wth the rebuttal. I stand by my original rating  and have not problem having the paper accepted.

**Limitations:**

A very bried discussion of the limitations in the Conclusion section.

**Quality:**

3

**Strengths And Weaknesses:**

Strengths:
The paper is well written, the proposed approach is simple and sound, and it gives good results, outperforming all baselines it is compared to.

Weaknesses:

The prediction of novel views appears to be a simple  transformer-based approach to depth map regression controlled by the choice of viewpoint, in the line of DUSt3R and related techniques, that regress point clouds to input views. This limits the novelty of the approach. The authors should justify the choice of  learning to render novel views, and aggregate the results, instread of directely regressing the depth of back-facing points, or any other proxy to the shape completion task.

---

> ### Author Rebuttal · Authors · 2025-07-31
>
> Thank you G7Ap for these great questions, we will modify the paper to reflect our answers below:
>
> Q1. With RaySt3R, we have shown that casting 3D shape completion as a novel view synthesis task enables leveraging recent insights from DUSt3R for object-centric tasks. With mask prediction and multi-view merging, we believe our method may be applied to other object-centric tasks such as 3D-complete tracking or panoptic segmentation. Besides the results, an important aspect is the ability to train on real-world data without GT meshes.
>
> Q2. In our setting, zero-shot transfer means we train RaySt3R only on synthetic data and evaluate the model on real-world scenes outside the training set, consisting of unseen objects, scene configurations, and noisy depth sensing input.
>
> Q3. The 3D visualizations show dense point clouds (not meshes), where colors are assigned based on each point's 3D location using a Turbo colormap.
>
> **Weaknesses**
>
> In addition to the weaknessess we addressed in answering the questions above, we note
>
> W1. **Justification of learning novel views vs other 3D-completion proxies**: Layered ray intersections (LaRI) is an example of an alternative proxy of 3D completion where effectively the back of an object is predicted. Benchmarking against LaRI and other volumetric approaches, the results suggest RaySt3R produces better completion results. RaySt3R can also be trained on any pair of (real-world) RGB-D views, all other baselines require access to the full scene geometry.
>
> **Limitations**: Thank you for pointing out the limited discussion in the conclusion, we commit to expanding the limitations and future work in the camera ready version. We will discuss the pros and cons of view-based multi-object completion. The advantages we see are the ability to query an arbitrary number of views to trade off quality and inference speed, the ability to supervise directly on real-world data without GT meshes, and empirically more accurate results compared to the volumentric baselines. View-based methods suffer from the common edge bleeding issue, and deciding where to sample novel views can be challenging. Finally, we will highlight RaySt3R's limitation to objects far beyond the current training dataset, such as transparent objects.

---

> > ### Comment · Reviewer_G7Ap · 2025-08-02
> >
> > Thanks for your rebuttal.
> >
> > - For Q1: I understand and appreciate the fact that you method gives good results. What I was asking about is what new insights and new ideas can be gained from the paper besides the fact that it works pretty well, since it appears to be a simple regression method. I would have the same question for DUST3R. What will the ML community learn from this work?
> >
> > - For Q2: my advice is to tone down the zero-shot claims, that, although current nowadays, seem a bit irrelevant here.
> >
> > - For Q3: thanks for clarifying.
> >
> > Thanks for committing to adding a discussion of limitations.

---

> > > ### Author Response · Authors · 2025-08-05
> > >
> > > Thank you for responding to our rebuttal, we answer below:
> > >
> > > Q1. Thanks for clarifying. In our view, DUSt3R showed that scaling up training data and introducing minimal inductive bias can enable new levels of performance. DUSt3R set a new milestone of feedforward 3D multi-view reconstruction, which layed an important foundation for a number of follow up works, including VGGT. Similarly, our work sets a new performance record for feed-forward single-view scene completion. This is achieved by contributing a novel large-scale dataset,  novel formulation of completion by querying view prediction, and enhanced multi-view consistency by confidence thresholding and fusion. We hope that together these contributions provide insights for exploring representations beyond 3D volumes in scene completion and other related tasks. This is important, since it demonstrates competitive results can be gained without GT meshes being used for the training process, which opens up the possibility of leveraging broader datasets for learning similar tasks in the future. Additionally, for the broader ML community, RaySt3R validates the effectiveness of scaling up training data, introducing minimal inductive bias, and scalable representations.
> > >
> > > Q2. Thank you for the suggestion. We will tone down and clearly state the zero-shot claims in the updated version.

---

> > > > ### Comment · Reviewer_G7Ap · 2025-08-05
> > > >
> > > > Thank you for your response.

---

> > > > > ### Author Response · Authors · 2025-08-06
> > > > >
> > > > > Thank you for your engagement in the rebuttal. As the author-reviewer discussions have been extended to August 8th, please let us know if you have any more questions that will help inform your decision.

---

### Official Review · Reviewer_bxUS · 2025-06-30

**Clarity:** 4
**Significance:** 3
**Originality:** 3
**Rating:** 5
**Confidence:** 3

**Summary:**

The authors proposed a method to recover complete 3D (without texture) from masked RGB-D image. They train a feed-forward  transformer to predict depth maps, object masks, and per-pixel confidence scores for novel views, and use information to help recover the missing part.

Contributions:
1. a good method to predict novel view information.
2. a new curated large-scale dataset that can help future research.
3. comprehensive experiment on different dataset and good performance.

Overall I think this model is easy to use and can contribute to the community.

**Questions:**

1. as shown in the teaser, the input depth may have some invalid or missing value in the corresponded RGB pixel, but they seems to share the same mask (line 87-88). Will this introduce noisy value to the network?

2. the model sample 22 views (line 194) in total to predict the full 3D. What will happen if sampling more or less views?

**Ethical Concerns:**

["NO or VERY MINOR ethics concerns only"]

**Final Justification:**

Thanks the authors to respond to my comments and the discussion from other reviewers. The authors test their pipeline in extra dataset (OctMAE) and shows good result. I read the comments from FXY9 and cWX6 and I think the threshold is not a determinant factor to reject this paper. Overall I think they propose an easy and straight-forward pipeline and I think the community can get some inspriation from this work.

**Limitations:**

yes

**Quality:**

4

**Strengths And Weaknesses:**

Strengths:

1. The writing is clear and easy to understand.

2. Good alignment. As shown in Fig. 3, although TRELLIS has better visual result, it's not as good aligned as the proposed model.

3. Good performance in both quantivive and qualitive result, as shown in the main paper and supplementary.

4. The paper did comprehensive ablations, like the range of hyper-parameters.

Weaknesses:

1. The model does not handle texture, which is also important for many real-world applications.

2. The objects in the experiment seems lies on a flat plane, and there are no heavy occlusion or stacking. It would be interesting to test model performance on those extreme cases.

3. The reconstructed object is not very smooth, and the reconstructed scene does not have individual instance segmentations.

---

> ### Author Rebuttal · Authors · 2025-07-31
>
> Thank you bxUS for these great questions, we will modify the paper to reflect our answers below:
>
> Q1. **Noisy masks**: Indeed the network may be challenged by noisy depth masks as present in the datasets. We try to mimic this situation in the data augmentations (Sec 5.3), and show it improves performance (Table 2).
>
> Q2. **Sampling views**: When sampling fewer views there is a higher chance of missing part of the geometry while inference will be faster. This is an application-specific tradeoff, e.g. when reconstructing only a single object fewer views may be sufficient.
>
> Below we reply to weaknesses raised in your review:
>
> W1. **Texture**: Geometry is the essential modality needed for a variety of robotic tasks such as grasping, 3D occupancy reasoning and motion planning, which is the focus of this work. In those applications, texture is not commonly used. However, we agree it would be an exciting future direction to also predict texture for broader applications such as digital twin creation.
>
> W2. **Non-planar predictions**: While we did not show these examples in the paper, the HOPE dataset contains several scenes where objects are placed in a non-planar fashion with heavy occlusions. For example, in HOPE's scene 3 objects are placed on the different parts of a chair (seat, handrest, headrest). In HOPE's scene 9, objects are placed on top of a cabinet as well as inside the drawers.
>
> W3. **Improving smoothness**: The lack of smoothness and occasional noisy points can be limited by setting a higher confidence threshold, sampling fewer views, or by fitting a TSDF to the depth map predictions. We will include an off-the-shelf TSDF fitting method in the open-source release.
>
> **Instance segmentation**: We agree individual instance segmentation would be a useful future direction, allowing users to export individual objects to a simulation environment. Given a modified dataset, a future version of RaySt3R may leverage ideas from PanSt3R [1] to also achieve instance segmentation.
>
> [1] PanSt3R: Multi-view Consistent Panoptic Segmentation, Lojze Zust, Yohann Cabon, Juliette Marrie, Leonid Antsfeld, Boris Chidlovskii, Jerome Revaud, Gabriela Csurka

---

> > ### Comment · Reviewer_bxUS · 2025-08-05
> >
> > Thanks for the rebuttal from the authors and the replied from other reviewer, especially the interaction with FXY9. I have known more details about the paper.

---

### Official Review · Reviewer_cWX6 · 2025-07-02

**Clarity:** 3
**Significance:** 3
**Originality:** 3
**Rating:** 5
**Confidence:** 4

**Summary:**

The paper addresses the task of reconstruction complete 3D objects/scenes from a single view RGB-D observation thereof.
To this end, the paper proposes to leverage transformer-based reconstruction network, which predicts depth maps, foreground masks and confidences for novel views. In order to complete an object/scene 22 novel views are sampled, and merged into a final point cloud prediction. The merging considers occlusions, confidence and mask predictions.

To train the model, the author propose a new large-scale synthetic dataset, which will be released to the public.

To evaluate the model the authors follow the evaluation procedure on OctMAE and copy their numbers and baselines. Additionally the authors add comparisons to LaRI, TRELLIS and Unique3D, all of which can only use RGB as input, while the proposed method sees RGB-D as input.

Finally, the authors ablate the importance of their merging scheme, and show ablations about ViT backbone architecture and training dataset options and usage of DINO-v2 features.

**Questions:**

- As mentioned above, I am mainly concerned about the current evaluations. The following comparisons are important for the paper:
    - Either train OctMAE on the newly proposed data, or train the proposed method on the original OctMAE training dataset using their augmentations. Otherwise, the experimental comparisons are completely inconclusive. (Not sure if the 226k GSO dataset ablation goes into a similar direction. If you the authors can elaborate, since the paper does not present all relevant details about it).
    - I believe the comparisons to TRELLIS and LaRI are not fair, since these models these models are designed for RGB inputs. For a fairer comparisons, the authors could for example train their method without the depth conditioning. Similarly, TRELLIS is mainly trained on single objects (for LaRI it depends which checkpoints the authors used), but evaluated on cluttered scenes. A fairer comparison that does not involve training TRELLIS on the proposed dataset, would be to evaluate single object reconstructions, e.g. using a subset of the current evaluation data, or datasets like CO3D.

- That being said, the formulation is certainly interesting and could prove valuable to the community in the future, e.g. LVSM follows a similar formulation is influential for NVS methods. Compared to other feed-forward reconstruction transformers, one distinction of LVSM is the lack of internal 3D representation and rendering formulation. Have the authors considered/experimented with adding an internal 3D representation to their model? After all it seems like an important design decision, which could be briefly discussed in the paper.

**Ethical Concerns:**

["NO or VERY MINOR ethics concerns only"]

**Final Justification:**

The authors have significantly improved the paper, by adding a crucial ablation experiment, additional clarifications, and comparisons on single objects. With these additions the experimental evaluation was raised above the acceptance threshold.

Overall, the paper presents significant value to the community (although the depth input is slightly niche), as it is the first method (to the best of my knowledge), which approach geometry reconstruction/completion using a view-based approach. While this results in less smooth geometry, e.g. compared to volumetric approaches, it is still a highly relevant research direction, especially, in times where view-based methods (LVSM, RayZer, GENIE3, VideoDiffusion models with camera control) are very successful for NVS. Therefore, the novelty aspect is also above acceptance threshold.

**Limitations:**

yes

**Quality:**

4

**Strengths And Weaknesses:**

**Strengths**
- (**S.a**) The paper casts 3d object/scene completion as a depth prediction task from novel views. This is a novel formulation, and especially interesting since methods like LVSM show great success with a similar formulation for novel-view RGB prediction.
- (**S.b**) The generated training dataset will likely benefit the community. I am not sure wether the effect of the new data is properly ablated though, since some information on the dataset ablations are missing (i.e. is 226k GSO, the data that OctMAE used?).
- (**S.c**) The paper is well-written and easy to follow.
- (**S.d**) The paper shows string experimental results, and, like OctMAE, successful manages the synthetic2real domain gap at inference time.


**Weaknesses**
- (**W.a**) OctMAE not trained on same dataset, so in the current state of the paper, it is not clear whether improvements are coming from the model/formulation or from the training data.
- (**W.b**) The paper does not mention LVSM, which follows a similar formulation for novel-view RGB synthesis. This should at least be referenced and/or discussed somewhere.
- (**W.c**) The comparisons to TRELLIS, LaRI etc. are not fair, because they don't get depth as input, and e.g. TRELLIS is used to model the distribution over single objects. The authors made no efforts of constructing fairer comparisons, like evaluating single object completion or training a version of their model without depth as input. In the current state, these added experiments merely present expectable results.

**Conclusion:**
The paper introduces an interesting paradigm for object/scene completion. However, the experimental validation of the presented ideas are lacking. Some comparisons are not fair because of vastly different training data, while other baselines receive only RGB instead of RGB-D inputs. Due to this lacking rigor of the evaluation, I am currently leaning towards rejection.

---

> ### Author Rebuttal · Authors · 2025-07-31
>
> Thank you cWX6 for these great questions, we will modify the paper to reflect our answers below:
>
> **(W.a) Comparison against OctMAE**:  	Following your suggestion, we trained RaySt3R only on the OctMAE dataset and with only the OctMAE augmentations. The table below shows RaySt3R trained on either dataset outperforms OctMAE (**bold** is best, _italics_ is second best).
>
> | Method             | Trained on | CD $\downarrow$ | F1 $\uparrow$ |
> | ------------------ | ---------- | --------:| -----------:|
> | **OctMAE**         | OctMAE     | 6.48     | 0.839       |
> | **Ours (RaySt3R)** | OctMAE (new) | _5.87_   |  _0.877_    |
> | **Ours (RaySt3R)** | RaySt3R    | **5.21** | **0.893**   |
> _Table 1. Model and training dataset comparision_
>
> While we did not have the resources to train OctMAE on our dataset, we believe that this experiment nevertheless shows the contribution of our method.
>
> In retrospect, our description of the datasets may have been unclear.
> Our dataset uses the exact same objects as OctMAE, and only 10\% more input images.  "1.1 million scenes", refers to the unique RGB-D input images; but the actual scenes are only 251k.  Also, "12 M views" includes both the input images and the depth maps used for supervision.  See table below.
>
> |  | **OctMAE** dataset | **RaySt3R** dataset |
> |---|---:|---:|
> | # RGB-D Input Images | 1 M | 1.1 M |
> | # Objects |   12 k   |    12 k  |
> | # Scenes (object configurations) | 25 k | 251 k |
> | Supervision | Sampled mesh | 11 M depth maps |
> _Table 2. Training dataset size comparision_
>
> **(W.b) LVSM** We agree LVSM is a relevant method with impressive performance, and commit to discussing it in detail in the final version of the paper. We cast 3D shape completion as a novel view synthesis problem which allows leveraging ideas from LVSM, DUSt3R and other recent works. A key difference between LVSM and RaySt3R is indeed the scene encoding, where we encode the query view extrinsics into the input view point map. The advantage of our approach is that we do not need full plucker coordinates in the query tokens, and instead only encode the intrinsics (HxWx2). A key advantage of the LVSM approach would be improved efficiency for many query views, as the encoder has to be run only once. We agree an ablation on this aspect would be useful for the community, but it goes beyond the compute resources available to us during the rebuttal.
>
> **(W.c) Comparisons vs TRELLIS and LaRI**
> Thank you for raising concerns about fair evaluation of the TRELLIS and LaRI baselines. First of all, our use case is to complete cluttered scenes based on a RGB-D image, which is a common while challenging setup in robotics. We believe the mentioned baselines are valuable in putting RaySt3R's results into perspective of recent and popular image to 3D pipelines, and help motivate depth conditioning for RaySt3R. We also made our best attempt to improve their results, by aligning the results of the baselines to the GT mesh with a brute force search + ICP, including a hyper parameter ablation (supp 1.2.1). RaySt3R outperforms the baselines, even when they are provided access to the GT mesh for alignment, which is unavailable to our method.
>
> However, following your suggestion, we additionally evaluate RaySt3R and baselines on a single object setting from YCB-V dataset. For each frame, we evaluate on the largest object with no occlusions which is often assumed by single-view RGB-based object reconstruction methods such as TRELLIS.
>
>
> | Method   | CD $\downarrow$ | F1 $\uparrow$ |
> | -------- | ----:| -----:|
> | TRELLIS  | 22.37 | 0.56  |
> | LaRI     | 13.75     |    0.51  |
> | OctMAE   |  8.34    |  0.74     |
> | Ours (RaySt3R)  | **4.06** | **0.91** |
> _Table 3. YCB-V single-object evaluation_
>
> **Clarifications**
> - **226k GSO in Table 2**: This ablation trains RaySt3R on the subset of our dataset curated from Foundation Pose, containing GSO objects only.

---

> > ### Comment · Reviewer_cWX6 · 2025-08-03
> >
> > I sincerely thank the reviewers for the detailed answers and additional experiments.
> >
> > Training the proposed method on the same dataset as OctMAE was a crucial missing part, and finally shows the proposed architecture has benefits compared to existing SotA methods.
> >
> > Regarding (**W.c**), I think there has been a missunderstanding. I did not refer to using the same input images and only evaluating on a single object. Instead I meant to use a subset of the dataset that has scenes with a single object only, or switching to a dataset like CO3D.
> > Currently, I still believe that the majority of comparisons are misleading to the readers, since:
> > 1. the majority of baselines is not train on cluttered scenes, but on single objects only
> > 2. the majority of baselines does not receive depth as input.
> >
> > At least the inputs to the methods should be more transparent, currently the comparisons are misleadingly presented, i.e. putting things in the same table suggests that the evaulation is fair, which it currently is not.
> >
> > I still beleive that one of the best ways to "help motivate depth conditioning for RaySt3R" (as you write in the rebuttal), would be to train RaySt3R without depth conditioning (e.g. using Plucker instead) and repeat the comparison. Would something like that be possible in general? Why did you not add it to the rebuttal? I guess for the rebuttal time is running short by now.
> >
> >
> > Overall, I am still not perfectly satisfied with the evluation of the method, but the added dataset-ablation certainly increases the paper's value. Before giving a final score, I would like to hear back from the authors once more, and hear the opinion from reviewer FXY9, since he seemed to critically think about the paper as well.

---

> > > ### Author Response · Authors · 2025-08-05
> > >
> > > Thank you cWX6 for responding to our rebuttal.
> > >
> > > **Wc.** The baselines we used in Table 1 of the paper submission fall into two groups:  the first 9, and the next 5 (separated by a horizontal line). Based on the reviewers' feedback, we will move these latter comparisons to a separate table so as to not confuse the reader.  In the camera-ready version, our main table will only compare with the first 9 methods.
> > >
> > > We respectfully disagree that the majority of comparisons are misleading to the readers, since the first 9 methods were treated the same as our method.  These first 9 methods were all trained on cluttered scenes, and they use depth as input.  As a result, we believe that our comparison with the first 9 methods is fair, since the training and input is the same as our method.  More specifically, all of these 9 baselines were trained on the OctMAE (cluttered) dataset and are conditioned on depth inputs---exactly like ours.  See table below.
> > >
> > > | Method | Trained on cluttered scenes | Receive depth as input | GT mesh for alignment |
> > > | ---- | :----: | :---: | :---: |
> > > | VoxFormer | YES | YES | no |
> > > | ShapeFormer | YES | YES | no |
> > > | MCC | YES | YES | no |
> > > | ConvONet | YES | YES | no |
> > > | POCO | YES | YES | no |
> > > | AICNet | YES | YES | no |
> > > | Minkowski | YES | YES | no |
> > > | OCNN | YES | YES | no |
> > > | OctMAE | YES | YES | no |
> > > | RaySt3R (ours) | YES | YES | no |
> > > | --- | --- | --- | --- |
> > > | LaRI | no | no | YES |
> > > | Unique3D | no | no | YES |
> > > | TRELLIS (w and w/o mask)  | no | no | YES |
> > > | SceneComplete | no | YES | no |
> > > | RaySt3R (ours) | YES | YES | no |
> > >
> > > We agree that our comparison with the remaining methods is not exactly apples-to-apples, because those methods do not receive depth as input and/or are not trained on cluttered scenes. We have indicated this difference using the extra horizontal line in Table 1 (of the paper).  As mentioned above, we will make this even more clear by moving them to a separate table and will clearly indicate baseline inputs in a table column, caption, and the paper text.
> > >
> > > Regarding the single-object evaluation, we tried to perform the experiment as we best understood the reviewer's  request.  To reiterate, in our experiment, we effectively create a single-object dataset out of the multi-object YCB-V dataset. We mask out the background except for a single selected object and only compare against the GT geometry of the single object. Importantly, note that this is a similar setup to the original TRELLIS evaluation in their paper, where images of objects without backgrounds are fed into the model.  Unfortunately, we did not have time to evaluate on the CO3D dataset, but we agree that this could be interesting for future research.
> > >
> > > The reviewer raises a very interesting suggestion of training without depth. However, we do not consider this within scope since our focus is on developing a method useful in contexts where depth is commonly available (such as robotics). For applications such as robotics, metric scale scene completion is critical, which is what we pursue in this project. With the ablations provided in the paper and rebuttal, we have shown both the RaySt3R architecture and dataset contribute to state-of-the-art results in this important task.

---

> > > > ### Comment · Reviewer_cWX6 · 2025-08-05
> > > >
> > > > Thanks you once more for the additional clarification.
> > > >
> > > > I apologize for my imprecise formualtion. I think having a single table with a horizontal line is the correct way (and better than two tables). I was merely referring to the new baselines, which are not coming from OctMAE.
> > > >
> > > > Thanks for clarifying the both the input images and geometry were masked. From your initial desccription, it seemed like you only mask the geometry, but not the input images.
> > > >
> > > > With this additonal input, I think most of my concerns have been addressed, and I am looking forword for the reviewer discussion in the next phase.

---

> > > > > ### Author Response · Authors · 2025-08-06
> > > > >
> > > > > Thank you for your response. We apologize for our imprecise formulation of the single-object experiment.
> > > > > As the author-reviewer discussions have been extended to August 8th, please let us know if you have any more questions that will help inform your decision.

---

### Note · Authors · 2025-08-12

We thank the reviewers for an engaging rebuttal. Throughout the rebuttal process, we have responded to all questions raised by reviewers, supported by comprehensive experimental results wherever possible. We did not receive further requests towards the end of the rebuttal. We hope that all reviewers' questions have been properly addressed. We have shown the following:

* We present a rigorous and fair comparison with existing methods. The majority of baselines are depth-conditioned and trained on cluttered scenes, with the same set of objects as RaySt3R. We additionally present results on a single-object dataset and will present the baselines more transparently. Our evaluation follows established evaluation protocols and baselines (e.g., OctMAE, ECCV '24), and additionally evaluates image to 3d baselines in our context.
* By suggestion of cWX6 and FX9, we train RaySt3R on the OctMAE dataset. The results suggest both our novel dataset and architecture contribute to state-of-the-art performance.
* We have provided a detailed explanation of noisy predictions and the confidence threshold, based on the suggestion from FXY9. We will release a TSDF-based post-processing module with the code release to create smooth meshes for applications where visual quality is a priority.

Summarizing, with RaySt3R we present a novel method and dataset that sets a new performance record in 3D completion of cluttered scenes. Our method improves on the state-of-the-art by up to 44% in real-world scenes. Given the experiments in the paper and rebuttal, we show that both the architecture and the new dataset are critical to outperforming existing methods.

---

### Decision · Program_Chairs · 2025-09-17

**Decision:**

Accept (poster)

**Comment:**

The final ratings are 2 borderline accepts, 2 accepts. The AC have read the reviews and rebuttal, and discussed the submission with the reviewers. The reviewers raised a number of points during the review phase including discussions of limitations, evaluation protocol, datasets, and comparisons. The authors were able to address many of these points during the rebuttal and discussion phases by including additional experiments on OctMAE dataset and clarification of view-based and volumetric representations in relation to the proposed method. While there are still some concerns remaining such as the confidence threshold raised by FXY9, the AC and reviewers reached a positive consensus. The AC recommends the authors to incorporate the feedback and suggestions provided by the reviewers, and the materials presented in the rebuttal (and additionally revisit the issues raised on confidence threshold), which would improve the next revision of the manuscript.